# Painful Cutaneous Electrical Stimulation vs. Heat Pain as Test Stimuli in Conditioned Pain Modulation [note 1]

**DOI:** 10.3390/brainsci10100684

**Published:** 2020-09-28

**Authors:** Elena Enax-Krumova, Ann-Christin Plaga, Kimberly Schmidt, Özüm S. Özgül, Lynn B. Eitner, Martin Tegenthoff, Oliver Höffken

**Affiliations:** 1Department of Neurology, Ruhr-University Bochum, Berufsgenossenschaftliches Universitätsklinikum Bergmannsheil gGmbH Bochum, 44789 Bochum, Germany; Ann.Plaga@web.de (A.-C.P.); Kimberly.Schmidt@ruhr-uni-bochum.de (K.S.); Oezuem.Oezguel@ruhr-uni-bochum.de (Ö.S.Ö.); martin.tegenthoff@rub.de (M.T.); Oliver.Hoeffken@rub.de (O.H.); 2Department of Pain Medicine, Ruhr-University Bochum, Berufsgenossenschaftliches Universitätsklinikum Bergmannsheil gGmbH Bochum, 44789 Bochum, Germany; Lynn.Eitner@ruhr-uni-bochum.de; 3Department of Neuropediatrics, University Children’s Hospital, Ruhr University, 44789 Bochum, Germany

**Keywords:** conditioned pain modulation, painful cutaneous electrical stimulation, heat pain, endogenous pain modulation, pain mechanisms

## Abstract

Different paradigms can assess the effect of conditioned pain modulation (CPM). The aim of the present study was to compare heat pain, as an often used test stimulus (TS), to painful cutaneous electrical stimulation (PCES), having the advantage of the additional recording of PCES-related evoked potentials. In 28 healthy subjects we applied heat and PCES at the dominant hand as test stimulus (TS) to compare the CPM-effect elicited by hand immersion into cold water (10 °C) as conditioning stimulus (CS). Subjects rated the pain intensity of TS at baseline, during and 5 min after CS application and additionally of CS, on a numerical rating scale (NRS) (0–100). The ‘early’ (during CS–before CS) and ‘late’ (after CS–before CS) CPM-effects were analyzed. Parallel to the PCES, the related evoked potentials were recorded via Cz to evaluate any changes in PCES-amplitudes. CS reduced significantly the pain intensity of both PCES and heat pain as TS. On a group level, the CPM-effect did not differ significantly between both paradigms. Both early and late CPM-effect based on PCES correlated significantly with the CS pain intensity (*r* = −0.630 and −0.503, respectively), whereas using heat pain the correlation was not significant. We found a significant reduction of PCES-amplitudes during CS, but this did not correlate with the PCES-induced pain intensity. Correlation with the CS painfulness (*r* = −0.464) did not achieve the significance level after Bonferroni correction. The extent of the CPM effects was similar in both testing paradigms at group level, despite intraindividual differences. Future studies should further elicit the exact mechanisms explaining the modality of these specific differences.

## 1. Introduction

Endogenous pain modulation seems to play an important role in the processing and transmission of nociceptive stimuli [1,2]. Conditioned Pain Modulation (CPM) is the surrogate model for its assessment in humans, based on the analgesic effect of a noxious conditioning stimulus (CS) on a noxious test stimulus (TS) [3]. The clinical relevance of the results obtained by CPM is of growing importance. The balance between inhibitory and facilitatory pathways has been reported to be impaired in several pain syndromes like fibromyalgia, irritable bowel syndrome or osteoarthritis [4,5] and to play a role in the development of postoperative pain and in the transition from acute to chronic pain [5]. Further, CPM can contribute to personalized pain medicine, identifying patients with a pronociceptive pain system, who seem to be more likely than non–pronociceptive patients to benefit from treatment with serotonin-norepinephrine reuptake inhibitors, as already demonstrated for patients with painful diabetic neuropathy and migraine [6,7].

CPM can be assessed by application of different noxious stimuli like heat, cold, electrical stimulation, laser or stress [8]. The recent recommendations on practice of CPM [9] highlighted the lack of uniform protocols for performing CPM and further data are needed to be able to compare different protocols, combine data from different studies and make conclusions on the clinical utilization of this test by meta-analyses and systematic reviews. It was proposed that a standard additional TS should be added to the one planned by the researcher. Furthermore, the use of sequential protocol was agreed, i.e., presentation of the TS immediately after, rather than only in parallel, to the CS provides a representation of endogenous pain modulation, free of biases such as distraction [9]. In previous studies we proposed the term “early” CPM-effect for the difference between the pain intensity of the TS before and during the CS, and “late” CPM-effect for the difference between the pain intensity of the TS before and after the conditioning [10,11].

Previous studies exploring cortical potentials (evoked by electrical or CO2 laser stimulation) and their modulation as test stimulus by a heterotopic noxious conditioning stimulus provided contrasting results regarding the nociceptive specificity of the effect [12,13,14,15,16,17,18,19], but objective readouts for the effect of conditioned pain modulation are still needed. Recently, we have introduced a novel CPM testing protocol based on a pinprick-like painful cutaneous electrical stimulation (PCES) using three custom-built concentric surface electrodes as TS and cold water bath as CS [20]. The advantage of this protocol is the concomitant recording of PCES-induced evoked cortical potentials as a measure to objectify CPM-effects by evaluating amplitude changes, which correlated significantly with the reported changes in subjective pain intensity.

The aim of the present study was to compare extent of the induced CPM-effects by painful cold water as CS using pinprick-like PCES as TS [20] with an established protocol using heat pain as TS [21], as it has been recently recommended to perform different protocols with the same subjects to increase the intraindividual reliability of different testing paradigms [9]. We further analyzed potential relations between the CPM-effects, based on both TS, and the pain intensity of the CS in order to evaluate any differences between both testing paradigms 

## 2. Materials and Methods

### 2.1. Subjects

The study protocol was in accordance with the latest version of the Declaration of Helsinki and was approved by the local ethics committee of the Faculty of Medicine, Ruhr-University Bochum, Germany (Reg. Nr. 17-6230). The study was performed from February to June 2018 in the Department of Neurology, University Hospital Bergmannsheil Bochum, Germany. Volunteers were recruited among students and their relatives after a detailed description of the assessment procedure and written informed consent.

Inclusion and exclusion criteria followed the recent recommendations for inclusion of healthy subjects into studies on sensory function [22]. Exclusion criteria were age <18 years, history of cardiovascular, gastroenterological, dermatological, metabolic, neurological, psychiatric or rheumatic diseases, current pain or upper respiratory tract infection, drug use, alcohol, or pain medication within the last 14 days before the assessment. Personal data such as age, sex, height and weight were documented before stimulation. Handedness was determined by the German version of the Edinburgh Handedness Inventory [23]. 

Relevant depression and anxiety symptoms were evaluated using the “Patient Health Questionnaire (PHQ4)” and all subjects with any of these symptoms were excluded. 

Other exclusion criteria were abnormalities indicating sensory impairment of the nociceptive system (especially Aδ- and C-fiber function). We excluded subjects with more than two values outside the 95% confidence interval for healthy subjects in the cold and warm detection thresholds (CDT, WDT) and mechanical pain threshold (MPT), which were performed according to the protocol of the German research network on neuropathic pain (DFNS) [24] and were compared to the published DFNS reference database [25].

### 2.2. Study Design

The study was designed as a randomized cross-over study and the subjects were assigned to either group A or B by the examiner. In group A, at first we assessed the CPM using PCES as TS, and afterwards using heat pain as TS, and in group B vice versa. 

### 2.3. Test Stimulus Calibration

#### 2.3.1. Painful Cutaneous Electrical Stimulation (PCES)

All tests were performed in an air-conditioned room, while the participants sat on a comfortable chair and were asked to avoid any movements to improve the quality of the evoked potentials. PCES was applied using three custom-built concentric surface electrodes, which consist of an external ring (anode), an isolating insert and a central cathode [26,27]. The electrodes were placed in a triangular formation (1.5–2 cm distance) at the dorsum of the dominant hand (supply area of the superficial radial nerve) and connected to a stimulator (Digitimer DS7A), as previously described [20]. We used a triangular formation of the electrodes to be able to induce a punctual pain intensity of 60 on a numerical rating scale (NRS 0–100, 0 = no pain and 100 = strongest pain imaginable) with less stimulation intensity. 

The electric stimulus, which was applied at the same time by all three electrodes, consisted of three monopolar square waves that last for 500 µs each with 5 ms interval in between. The duration of the monopolar square waves was modified compared to the initially published protocol, to be able to achieve punctual pain intensity of 60 on a NRS (0–100) with less stimulation intensity and, thus, activate less non-nociceptive fibers. Four electric stimuli formed a block with an interstimulus interval of 4–6 s. The inter-block interval of 12 s was used to let the subjects rate the overall PCES-pain intensity of the last four stimuli on the NRS 0–100). One session consisted of five blocks (and 20 stimuli) and lasted for 96.2 s (see Figure 1), as previously described [20].

The intensity of the TS was determined by increasing the electric current in steps of 0.2 mA until the subjects reported a reliable PCES-induced pain of NRS 60. Each stimulus evokes a cortical potential which was recorded over one electrode placed at “Cz” following the international 10–20 EEG system, referred to linked earlobes (A1 and A2) and stored for later offline analysis (Brain Amp, Brain Products, Germany; bandwidth 1 Hz–1 kHz, sampling rate: 5 kHz). Impedances were kept below 5 kΩ. With Brain Vision Analyzer v. 1.05 (Brain Products, Germany) the painful cutaneous electrical stimulation (PCES-EPs) were analyzed offline in epochs of 1000 ms, including the time period of 200 ms before to 800 ms after application of the stimulus and were averaged after application of a notch filter. We rejected the first stimulus to avoid contamination by movement artefacts [27,28,29]. We determined for each averaged potential N1- and P1-peaks manually by visual recognition in order to calculate the N1-P1 peak-to-peak amplitudes. The time frame chosen to select the peaks for the N1-peak was 80–200 ms, and the P1-peak was determined as the following most positive peak within a time frame of 180–200 ms, as in previous studies [28,30].

#### 2.3.2. Tonic Heat

Heat stimuli were applied with a thermal sensory testing device (TSA 2001-II, MEDOC, Israel, CoVAS software, version 3.20) using a thermode with a contact area of 30 × 30 mm and a stimulus ramp of 4 °C/s [21]. First, three heat stimuli (45, 46 and 47 °C) were applied for 7 s to the dominant volar forearm. Between the application of the stimuli a pause of 35 s was used to attach the thermode 30 mm proximally after the first stimulus and 30 mm distally from the first position after the second stimulus to avoid thermal sensitization [31]. The subjects rated the pain intensity on the NRS (0–100). The intensity of the TS was defined as the heat stimulus temperature rated 60 ± 10 on the NRS (0–100). In case the standard temperatures (45–47 °C) did not reach the required pain intensity, two lower (43 and 44 °C) and two higher (48 and 49 °C) temperatures were used. 

### 2.4. Conditioning Stimulus

As CS the non-dominant hand was immersed up to the wrist in a cold water bath kept at 10 °C for 60 s for heat pain-CPM and 126.2 s for PCES-CPM (details see below). The temperature was measured by a calibrated digital thermometer (0–100 °C) with an accuracy of ±1 °C. The CS was not individually calibrated, in accordance with previous studies, confirming a sufficient CPM-effect [10,21,32,33]. The CS was applied for 30 s prior to additional application of the TS. 

### 2.5. CPM-Procedure

The individually determined TS, corresponding to a pain intensity of 60 ± 10 on the NRS (0–100) was applied during the CPM assessment as TS_baseline_, TS_during_ and TS_after_ according to the previously published protocol by Granot et al. [21]. Measurements were performed at the dorsum of the dominant hand, so that test areas for both testing paradigms matched. Each time, the subjects were asked to rate the TS pain intensity. Five minutes after the first TS (TS_baseline_), subjects were asked to put their hand into the cold water with spread fingers and without touching the bottom or the walls of the container. After 30 s immersion of the hand in the cold water bath, the TS was applied simultaneously (TS_during_) again. Subjects rated the pain intensity of the TS and additionally the pain intensity of the CS. Finally, 5 min after termination of the CS, the TS was applied again (TS_after_) and its intensity was rated.

Lower negative values after calculation of the CPM-effect (see below) indicated greater endogenous pain inhibition [9]. 

#### 2.5.1. CPM-Paradigm Based on PCES

At baseline, PCES as TS was applied by 20 test stimuli and subjects rated the pain intensity after every four stimuli on the NRS (0–100) as previously described (Figure 1B) [20]. During CS the subjects put their non-dominant hand into the cold water bath and after 30 s the PCES was applied again for 96.2 s. The participants rated the PCES-induced pain and the cold water pain after every block consisting of four stimuli and then were allowed to remove the non-dominant hand from the cold water. Thus, in this testing paradigm, the CS was applied for 126.2 s. Five minutes after removing the CS, PCES was applied again and pain intensity was determined equally to the baseline assessment.

#### 2.5.2. CPM-Paradigm Based on Heat Pain 

At baseline, the TS temperature was applied for 30 s, and after 10 s, 20 s and 30 s the subjects rated the heat pain intensity on the NRS (0–100). During CS [21] the subjects put their non-dominant hand into the cold water bath and after 30 s the TS temperature was applied again for 30 s. The participants rated the heat pain and the cold water pain 10 s, 20 s and 30 s after the TS was applied and the CS-induced pain additionally 30 s after immersion into the cold water bath. Then they were allowed to remove the non-dominant hand from the cold water. After CS, the TS temperature was applied and pain intensity was [21] determined, as in the baseline assessment.

#### 2.5.3. Calculation of the Early CPM-Effect

For the calculation of the early CPM-effect the means of the pain ratings for “baseline” and “during CS” were determined. For PCES-CPM the mean consists of five pain ratings for each session and for Heat-Pain-CPM the mean consists of three pain ratings for each session. The early CPM-effect was calculated as follows: Early CPM-effect = mean of the pain ratings TS_during_ − mean of the pain ratings TS_baseline_

#### 2.5.4. Calculation of the Late CPM-Effect

For the calculation of the late CPM-effect the means of the pain ratings for “baseline” and “after CS” was determined. For PCES-CPM the mean consists of five pain ratings for each session and for Heat Pain-CPM the mean consists of three pain ratings for each session. The late CPM-effect is calculated as follows: Late CPM-effect = mean of the pain ratings TS_after_ − mean of the pain ratings TS_baseline_

#### 2.5.5. CPM-Effect Based on the PCES-Amplitude

For the PCES-CPM testing paradigm, we calculated a CPM-effect based on changes in the PCES-induced N1/P1-amplitudes. The early and late CPM-effect based on the PCES-amplitude were calculated as follows, respectively:Early PCES-amplitude effect = N1-P1 amplitude_during_ / N1-P1 amplitude_baseline_
Late PCES-amplitude effect = N1-P1 amplitude_after_ / N1-P1 amplitude_baseline_

### 2.6. Statistical Analysis 

All analyses were conducted using SPSS (Statistical Package for the Social Sciences) version 26 (SPSS, Chicago, IL, USA). Descriptive statistics are presented as means and standard deviations. For the comparison of the mean NRS values (dependent variable) ANOVA for repeated measures was performed with within-subjects factors “session” (“baseline”, “during CS”, “after CS”) and “testing paradigm” (PCES-CPM and Heat Pain-CPM) and between-subject factor “randomization group” (randomization group A or B). For the comparison of the N1–P1 peak-to-peak amplitudes (dependent variable) of each session (within-subject factors), ANOVA for repeated measures was performed with between-subject factor “randomization group” (A or B). Pearson correlation analysis was performed to analyze any possible correlation between the early and late CPM-effects with each other as well as with the rated pain intensity of the CS for both testing paradigm separately, and additionally only for PCES-CPM between the CPM-effects based on the PCES-amplitudes and the pain ratings for TS and CS. Due to the exploratory character of this analysis, we have applied correction for multiple comparisons, using familywise Bonferoni correction, resulting in a significance level of *p* = 0.006 for this part of the analysis. Two-tailed paired *t*-tests were used to evaluate differences between the first and the last rating of each session (baseline, during CS, after CS), to evaluate differences between the values during CS and after CS compared to the values at baseline (familywise Bonferroni correction for multiple comparisons, resulting in a significance value of *p* = 0.006 for that part of the analysis) and to evaluate differences between the CPM-effect magnitude of both testing paradigms. 

## 3. Results

### 3.1. Subjects

One subject was excluded due to an abnormal cold-detection threshold (hypesthesia), warm-detection threshold (hypesthesia) and mechanical pain threshold (hyperalgesia) in the quantitative sensory testing (QST). A further two subjects were excluded because the evoked potentials could not be utilized and a further two were excluded due to technical error during the PCES (Figure 2).

Thus, 28 subjects (13 female and 15 male; age 31.4 ± 12.1 years, range 20–56 years, 26 right-handed and 2 left-handed) were included in the study analysis. 13 subjects were allocated to group A and 15 to group B. According to the Body-Mass-Index (BMI), 15 subjects had normal weight, three were overweight, seven were obese and three were underweight. 

### 3.2. Stimulation Intensities for Both CPM Paradigm

Assessment of the CPM-effect using PCES and heat as TS was possible in all subjects and none of them had to quit due to intolerably painful stimuli. 

The mean stimulus intensity to induce a PCES-induced pain of NRS 60 was 6.4 ± 5.5 mA (range: 1.6–27 mA). All subjects tolerated the cold water bath (10.1 ± 0.2 °C) as CS during the CPM based on PCES-induced pain for the necessary time.

The mean test stimulus temperature to induce a heat pain of NRS 60 ± 10 was 46.2 ± 1.6 °C. For 71% (*n* = 20) we used the standard temperature (45–47 °C), for 14% (*n* = 4) each we used the higher (48–49 °C) or lower (43–44 °C) temperature. All participants tolerated the cold water bath (10.1 ± 0.2 °C) during the CPM based on heat pain for the necessary time.

### 3.3. Comparison of the Pain Ratings between Both Testing Paradigms

The mean pain ratings of the TS during the CPM testing paradigm based on PCES were 65.5 ± 12.1 at baseline, 49.5 ± 13.5 during CS application and 57.3 ± 11.2 after CS. The mean pain ratings of the TS during the CPM testing paradigm based on heat pain were 50.2 ± 17.4 at baseline, 36.7 ± 19 during CS application and 42.7 ± 21 after CS (Figure 3A). 

ANOVA for repeated measures showed significant differences for the mean pain ratings between each session, i.e., prior to, during and after CS. The mean pain ratings were significantly lower both during and after CS compared to the baseline values for both testing paradigms (all *p* < 0.001, significant also after familywise Bonferroni correction). However, the mean pain ratings differed significantly between both testing paradigms. Pain ratings induced by heat pain were constantly lower than the PCES-induced pain ratings. In contrast, none of the interactions between the within-subject “time” and the within-subject “testing paradigm” and the between-subject factor “randomization group” showed any significant influence (Table 1). 

To further explore the reasons for the significant difference between the pain ratings of both testing paradigms we analyzed the change in pain ratings during TS application of the same intensity, which revealed an increase in the PCES-induced pain ratings over the repeated queries at baseline and during CS application (Table 2). In contrast, the heat pain ratings decreased over the repeated queries during TS application at baseline, during CS application and after CS, respectively (Table 2). In contrast to the comparison of the mean pain ratings, there was no significant difference between the first pain ratings of both TS (F _1;56_ = 1.552, *p* = 0.218).

ANOVA for repeated measures showed significant differences between the first pain ratings at baseline, during CS and after CS (F _2; 104_ = 42.644, *p* < 0.001). Neither the between-subject factors “testing paradigm” (F _2; 104_ = 3.029, *p* = 0.053) nor “randomization group” (F _2; 104_ = 1.776, *p* = 0.174) or their interaction (F _2; 104_ = 1.296, *p* = 0.174) showed a significant influence.

### 3.4. Comparison of the CPM-Effects Based on Pain Ratings between Both Testing Paradigms

Neither the early (*p* = 0.474) nor the late (*p* = 0.804) CPM-effect differed significantly in their magnitude between both testing paradigms (Figure 3B).

The number of subjects with a CPM-effect <0 did not differ between both testing paradigms, neither for the early nor for the late CPM-effect (Table 3). All participants showed a CPM-effect < 0 on the NRS (0–100) during either of the two testing paradigms. Using PCES as TS in one subject the test paradigm could elicit neither an early nor a late CPM-effect <0, and regarding the late CPM-effect this was the case in a further four subjects. Using heat as TS the test paradigm could elicit neither an early nor a late CPM-effect < 0 in two subjects and in a further two subjects this was the case for either of the CPM-effects.

However, regarding the intraindividual comparison neither the early effects (Figure 4A) nor the late effects (Figure 4B) based on both paradigms correlated significantly with each other.

### 3.5. Relation between the Magnitude of the CPM-Effect and CS

For PCES-CPM the early (*r* = −0.630, *p* < 0.001) and the late CPM-effect (*r* = −0.503, *p* = 0.006) correlated significantly with the pain intensity induced by cold water (Figure 5A,B). For the CPM testing paradigm with heat as TS neither the early nor the late CPM-effect correlated significantly with the pain induced by the cold water (Figure 5C,D).

The mean pain rating of the cold water bath was higher during the PCES-CPM (52.1 ± 22.7) than during the CPM with heat as TS (45.7 ± 22.2, *p* = 0.002). There were no significant differences regarding the CS pain intensity depending on the sequence of the CPM-testing paradigm.

### 3.6. CPM-Effect Based on PCES-Amplitudes

The N1-P1 amplitude was 43.2 ± 16 µV at baseline, 34.2 ± 14.3 µV during CS application and 42.4 ± 19.7 µV after CS (Figure 6). ANOVA for repeated measures showed significant differences of the N1P1-amplitudes for each session (F _2; 52_ = 13.061, *p* < 0.001). The between-subject factor group showed no impact.

The relative changes of the PCES-amplitudes corresponded to −18.7 ± 21.1% for the early CPM effect and 0.57 ± 23.5% for the late CPM effect.

Neither the early nor the late CPM-effect based on PCES-amplitudes correlated with the corresponding CPM-effect based on the pain ratings. The correlation between the early CPM-effect based on PCES-amplitudes and the CS pain intensity (*r* = −0.464, *p* = 0.010) did not achieve significance level after Bonferroni correction. There was also no significant correlation between the late CPM-effect based on PCES-amplitudes and CS pain intensity (*r* = −0.298, n.s.).

## 4. Discussion

The aim of the present study was to compare the CPM-effect elicited by an established paradigm with heat as TS and cold water bath as CS to a newly proposed paradigm using the same CS and PCES as TS. In summary, in healthy subjects the CPM-effect based on both TSs was significant, and group comparison did not reveal any differences between both testing paradigms. Interestingly, on an intraindividual level, the CPM-effects based on both TSs did not correlate. Further, subjects rated the TS intensity of the repeated PCES on average higher than the repeated heat stimuli, while the rating of the first stimulus of both modalities was similar.

Notably, the specificity for nociceptive fibers of PCES is still controversially debated [34,35]. The aim of the present study was to further elucidate the understanding of this recently introduced CPM model using PCES as TS with additionally recording of PCES evoked cortical potentials [20,36].

### 4.1. Extend and Duration of the CPM-effect

During CS the PCES- and heat-induced pain was reduced on average by 23 ± 18% and 29 ± 25%, respectively. The TS pain intensity after CS application is regarded as displaying the pain modulation, free of biases such as distraction [9]. However, there are few data on the persistence of CPM-effect after termination of the CS [10,37,38,39,40]. In the present study the late CPM-effect, i.e., endogenous inhibition lasting for at least 5 min after CS termination, was smaller than the early CPM-effect during CS application, in line with previous studies. The magnitude of the early CPM-effect induced by heat as TS was similar to our previous reliability study [10], though the range in the existing literature is quite high and depends also on the subjects’ age [41]. Interestingly, the pain reduction induced by PCES as TS in the current study was lower compared to our previous two studies, which was also paralleled by smaller changes of the N1-P1 amplitudes of the PCES-evoked potentials [20,36] A study demonstrated that the CPM-effect depends on the perceived level of the CS pain when it is altered by cognitive manipulation, rather than solely on its physical intensity [42]. One possible explanation for the difference of the PCES-induced outcome parameters in the present analysis could be the lower pain intensity of the CS compared to the previous two studies [20,36], although the cold water bath had the same temperature of 10 °C in all three studies.

In the present study the CPM-effect correlated with the CS pain intensity only in the case of PCES-induced pain as TS, in line with our previous study [20], underlining the validity of this finding. Interestingly, there was no such correlation between the CS and heat pain as TS, similarly to a study which also applied heat as TS, demonstrating that the magnitude of the CPM-effect based on heat-induced pain intensity did not depend on the perceived CS pain intensity which was influenced by cognitive manipulation, but only on the CS temperature [43]. Notably, the duration of the CS application differed between both paradigms, as the duration needed for recording of PCES-induced cortical potentials during CS was longer that in the other test paradigm (60 s vs. 126.2 s). Thus, the correlation between CPM-effect and CS painfulness may depend on the duration of the CS or the modality of TS.

In accordance with our previous studies [20,36], the amplitudes of the PCES-induced evoked potentials were significantly reduced both during and after CS application, though, the reduction did not correlate with the changes in TS pain intensity. The latter is in contrast to our first study [20], but in line with following studies we performed using the same CPM testing paradigm [36]. This might be due to the less pronounced mean CPM-effect in the current study compared to the previous [20]. Notably, the CPM-effect had a similar magnitude in both previous studies [20,36], but the correlation between pain intensity and the amplitudes of the PCES-induced evoked potentials was significant only in one of them.

### 4.2. Defining Responders and Non-Responders

The consideration of measurement error in the calculation of a potentially clinically meaningful effect has been recommended as a statistically robust method for the interpretation of score changes [44,45]. A definition has been proposed of a meaningful CPM effect as a percentage change from baseline greater than the inherent measurement error [46]. However, this approach requires reliability studies, which are still to be published for the newly introduced CPM testing paradigm with PCES as TS. We have previously reported reliability measures for heat as TS, but for a period of 24–72 h [10]. Due to missing reference data enabling the calculation of measurement error for both applied CPM testing paradigms, we have defined CPM responders as subjects with CPM-effect >0. The PCES-induced pain intensity decreased in 96% of the subjects during CS and was still lower than the corresponding value at baseline in 79% of subjects 5 min after CS termination, in contrast to other published protocols, which were not able to induce a CPM-effect in a substantial amount of subjects [44]. When the CPM-effect was calculated based on the persisting changes of pain intensity 5 min after the CS, the reduction of the PCES-induced and the heat-induced pain was less pronounced and less subjects reported a CPM-effect <0, confirming previous data of a short lasting CPM-effect [10,47,48,49]. However, 5 min after CS termination the pain reduction compared to baseline was still significant for both testing paradigms, indicating that the CPM-effect diminishes over time, but is still at least partially present 5 min after CS termination. Further studies delineating the exact time course of the CPM-effect are needed.

### 4.3. Modality Specific Effects

Comparing the reduction in pain intensity of both TSs, i.e., heat and PCES, the average magnitude of the CPM effect did not differ between both testing paradigms. We also did not find any significant carry-over effects in our cross-over design. However, on the intraindividual level, neither the early nor the late effects correlated significantly between both testing paradigms. Although it has been recently recommended to enrich studies and perform different CPM protocols with the same subjects [50], only a few studies have compared the magnitude of the CPM-effect between different CS [51,52,53] or different combinations of CS and TS [54,55]. Significantly different CPM magnitudes were evoked in the same individual depending on the nature of the CS, even when the TS-induced and the CS-induced pain intensity were similar [51]. The cold pressure pain was shown to evoke the largest CPM effect in combination with the pressure pain thresholds as TS, having the best intra- und inter-individual reliability measures [52,55]. When the CPM effect was compared between different TS (electrical, heat, handled and cuff pressure pain threshold), baseline pain ratings differed significantly between the CPM protocols with heat pain thresholds and pressure pain thresholds as TS [54].

In our study the pain intensity at baseline after the first stimulus application was comparable for both TSs, although the mean rating after repeated application of the same intensity differed significantly. While the PCES-induced pain ratings increased slightly though not significantly, the heat pain ratings decreased significantly over the repeated queries. The reduction of pain intensity after repeated heat application is in line with previous studies, which reported habituation effects for heat pain threshold with a significant increase between the first and sixth stimuli [56]) and a rapid decrease of pain perception within the first stimuli [57]. Both are explained by the peripheral fatigue of the receptors. In contrast, the slight increase of PCES-induced pain intensity cannot be explained solely by peripheral mechanisms. In contrast, repeated PCES application with 2 min between the blocks led to a reduction of mean pain intensity, which was paralleled by a reduction of cortical evoked potentials [36], and central habituation processes were hypothesized. However, the slight increase in pain intensity in our study was observed after short-term repeated application. As the electrodes were stuck in the same position, this might be due to temporal and spatial summation of repeated electrical stimuli due to summation of C- and/or A-δ-fiber evoked responses, which generate a progressive increase in action potential discharge in the second-order neurons resulting in central sensitization [58,59]. Regardless of the underlying mechanisms, this finding might be relevant to a comparison between different CPM paradigms, as repeated TS application at baseline is generally used for baseline assessment but might have different implications depending on the TS used.

### 4.4. Strengths and Limitations

One limitation of our study is the fact that we did not included a control condition using a CS with room temperature water due to practical reasons. Including such a control condition for both parts of the cross-over study design would have resulted in a much longer experiment with the need for a much higher number of study participants. Such a control condition might have addressed habituation as a potential confounder of repeated heat testing. Previous studies have already performed such comparisons and displayed that painless water of 33 °C as CS, but also CS with temperature of 15 °C, 18 °C and 44 °C, were not able to induce a significant CPM effect based on contact heat pain as TS [21]. We have also previously compared the CPM effect based on PCES-induced pain as, using a control condition with CS of 24 °C and could not detect a significant CPM-effect for the reported pain intensity or for neurophysiological readouts in the control condition [20]. Further, we did not record contact heat pain evoked potentials. Specific devices for recording contact heat pain evoked potentials are cost-intensive and to our knowledge CPM testing paradigms using contact heat evoked potentials as TS are not common. Future studies are needed to compare modality specific processing of CPM, e.g., based on cortical potentials evoked by PCES, pinprick, laser, contact cold or contact heat.

A strength of the present study is that the recently proposed protocol using PCES as TS and painful cold water bath as CS is able to induce a CPM effect of similar intensity, based on subjective pain ratings, as using a common protocol with contact heat as TS, at least at a group level. Using concentric electrodes to elicit PCES-EP enables a reliable judgment of the intensity of brief, distinct and well-tolerable pinprick-like pain in small skin areas, and also enables a quick adjustment to target pain intensity by modifying current intensity [20]. According to the present protocol, we recorded cortical evoked potentials over Cz. A recent study on cortical mapping of painful electrical stimulation by quantitative electroencephalography confirmed this location as suitable for capturing a reproducible cortical neural response after painful electrical stimulation [12]. This underlines the validity of our protocol as feasible. Thus, a major advantage of the present CPM protocol using PCES as TS is the possibility of recording event-related potentials and appraising the change in their amplitude as an objective measure of human CPM using cost-effective common devices for neurophysiological examination, which makes it potentially available for a majority of clinicians.

## 5. Conclusions

In conclusion, the recently proposed CPM testing paradigm using the cold water bath CS and PCES as TS induces at group-level a CPM effect of similar magnitude compared to that using the same CS and contact heat as TS, though intraindividual differences were apparent and the CPM effect correlated with the perceived pain intensity of the CS. Further studies should elucidate the reasons which explain the different modality-specific intraindividual responses based on both different TS and exploring differences in the test-re-test reliability between both testing paradigms.

## Figures and Tables

**Figure 1 brainsci-10-00684-f001:**
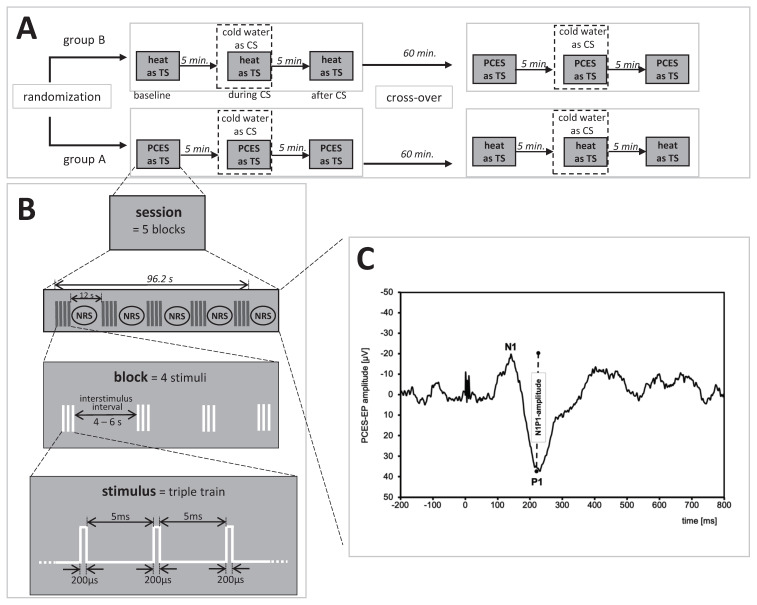
Study design. (**A**) Timeline of experimental procedure, (**B**) paradigm for electrical stimulation and (**C**) evoked potential after painful cutaneous electrical stimulation (PCES-EP) with N1 and P1 peaks recorded over Cz of one subject. TS: test stimulus; CS: conditioning stimulus; NRS: assessment of the pain intensity during painful cutaneous stimulation on the numerical rating scale.

**Figure 2 brainsci-10-00684-f002:**
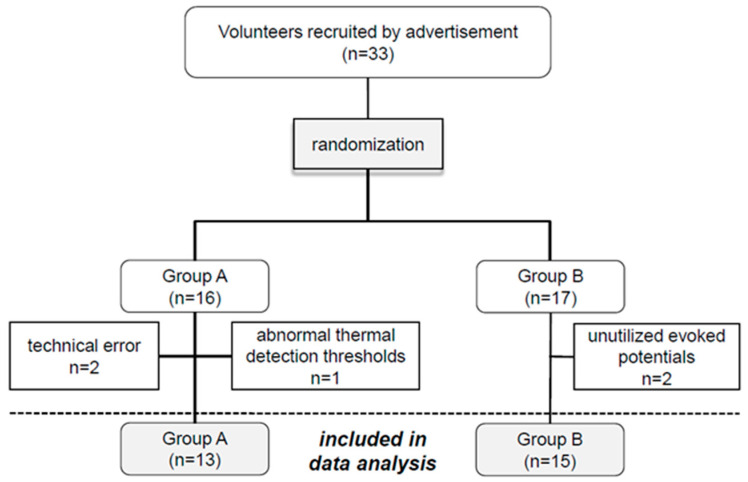
Randomization and exclusion of study participants.

**Figure 3 brainsci-10-00684-f003:**
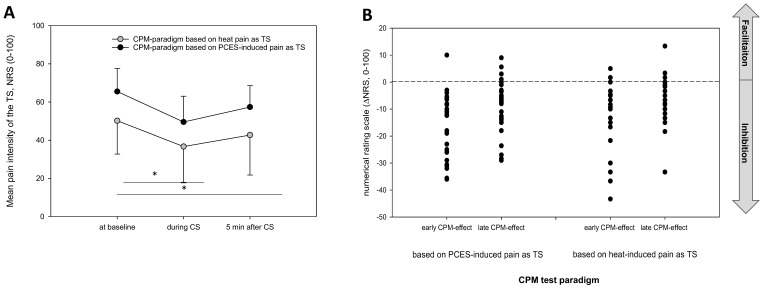
(**A**) Comparison of the pain intensity between the conditioned pain modulation paradigm using painful cutaneous electrical stimulation (PCES) and heat pain as test stimulus (mean ± SD; numerical rating scale (NRS) 0–100, 0 = no pain and 100 = strongest pain imaginable). * significance *p*-value below the level of significance (**B**) Individual early (difference between values during conditioning stimulus (CS) and values at baseline) and late conditioned pain modulation (CPM)-effects (difference between values after CS and values at baseline) based on both testing paradigms, negative values indicating endogenous pain inhibition.

**Figure 4 brainsci-10-00684-f004:**
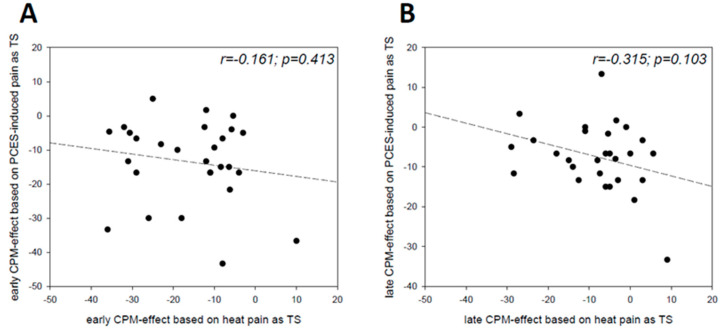
(**A**) Scatter plot of the early CPM-effect based on painful cutaneous electrical stimulation (PCES) as test stimulus (x-axis) and based on heat pain as test stimulus (y-axis). (**B**) Scatter plot of the late CPM-effect based on painful cutaneous electrical stimulation (PCES) as test stimulus (x-axis) and based on heat pain as test stimulus (y-axis).

**Figure 5 brainsci-10-00684-f005:**
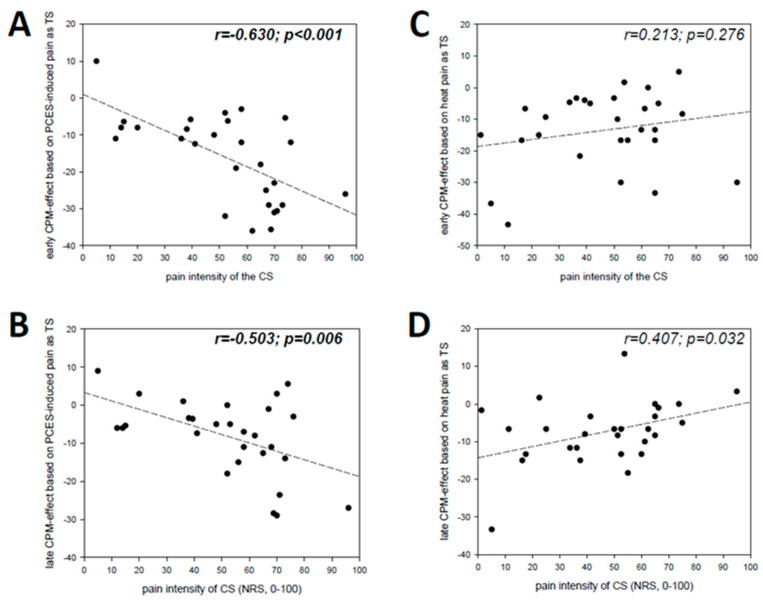
Scatter plot displaying correlations between the pain intensity of the conditioning stimulus with the (**A**) early and (**B**) late CPM-effect based PCES-induced pain as TS as well as with the (**C**) early and (**D**) late CPM-effect based on heat pain as TS.

**Figure 6 brainsci-10-00684-f006:**
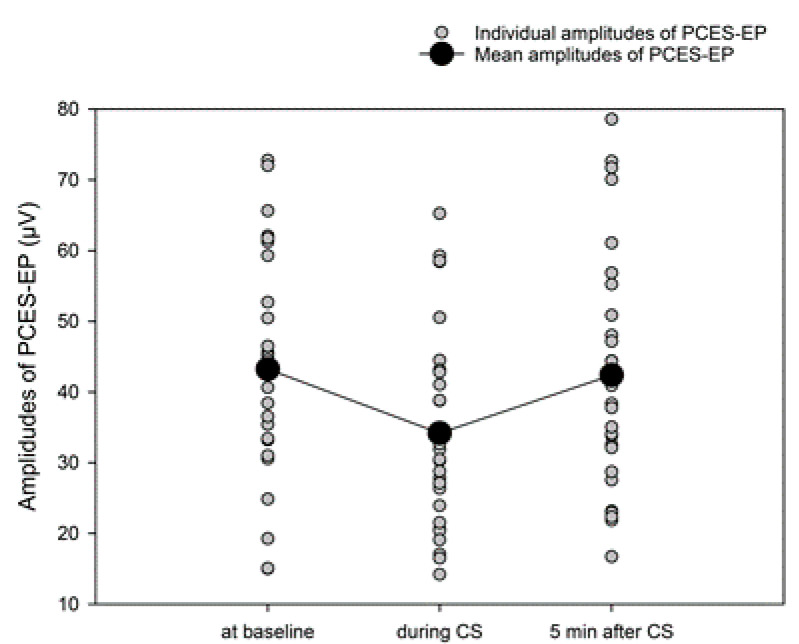
Amplitudes of cortical potentials evoked by PCES-EP at baseline, during CS and five minutes later.

**Table 1 brainsci-10-00684-t001:** Repeated measures ANOVA for PCES-induced and heat pain intensity with within factors “time” (“at baseline”, “during CS”, and “after CS”) and “testing paradigm” (“PCES” and “heat”) and between-subject factor (“randomisation group”).

Factor	F-Value	Significance	Partial *η*^2^
time	F 2;52 = 62.931	***p* < 0.001**	0.708
time * randomisation group	F 2;52 = 1.295	*p* = 0.283	0.047
testing paradigm	F 1;26 = 14.744	***p* = 0.001**	0.362
testing paradigm * randomisation group	F 1;26 = 3.931	*p* = 0.058	0.131
time * testing paradigm	F 2;52 = 0.373	*p* = 0.690	0.014
time * testing paradigm * randomisation group	F 2;52 = 0.558	*p* = 0.576	0.021

Bold: *p*-value below the level of significance. * Interaction between both factors tested by ANOVA.

**Table 2 brainsci-10-00684-t002:** Comparison of the first and last pain ratings of the test stimulus.

	PCES-Induced Pain			Heat Pain		
	First Pain Rating	Last Pain Rating	*p*-Value	First Pain Rating	Last Pain Rating	*p*-Value
**Baseline**	62.1 ± 14.4	67.4 ± 14.5	0.080	61.6 ± 13.9	43.2 ± 23.5	**<0.001**
**During CS**	46.9 ± 13.2	51.1 ± 15.2	0.076	43.4 ± 18.7	31.0 ± 21.2	**<0.001**
**After CS**	55.1 ± 13.8	57.9 ± 12.8	0.241	49.8 ± 19.7	37.0 ± 22.8	**<0.001**

Bold: *p*-value below the level of significance.

**Table 3 brainsci-10-00684-t003:** Conditioned pain modulation based on PCES and heat as test stimulus (TS).

	CPM Testing Paradigm Based on PCES		CPM Testing Paradigm Based on Heat Pain	
	Early CPM-Effect	Late CPM-Effect	Early CPM-Effect	Late CPM-Efect
**absolute number of responders (CPM-effect >0)**	27	22	25	23
**relative number [%] of responders**	96	79	89	82
**absolute magnitude of the CPM-effect**	−16.0 ± 11.8	−8.17 ± 9.99	−13.5 ± 12.0	−7.5 ± 8.4
**relative magnitude [%] of the CPM-effect**	−22.7 ± 17.9	−11.5 ± 14.3	−29.2 ± 24.8	−19.3 ± 23.4

CPM: conditioned pain modulation, PCES: painful cutaneous electrical stimulation; early CPM-effect: change of pain intensity during conditioning stimulus compared to baseline; late CPM-effect: change of pain intensity 5 min after application of the conditioning stimulus compared to baseline.

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
