# Peer review of "Painful Cutaneous Electrical Stimulation vs. Heat Pain as Test Stimuli in Conditioned Pain Modulation"

_brainsci, 2020, doi:10.3390/brainsci10100684_

Round 1

Reviewer 1 Report

Enax-Krumova/Plaga et al. present human subject data showing that painful cutaneous electrical stimulation (PCES) can be used as a test stimulus for measuring the effect of conditioned pain modulation (CPM). The authors also used traditional heat pain as a test stimulus for comparison. Cold water (10 degrees Celcius) was used as the conditioning stimulus(CS) to evoke CPM. The authors tested for CPM at both an early and late time point, which corresponded to during and after the conditioned stimulus, respectively. While CPM was observed using both heat pain and PCES, CS pain intensity on correlated with CPM-effect using PCES as the test stimulus. In addition, cortical activity evoked by PCES was reduced the CS. While this study presents some interesting differences in the mechanisms of CPM across different pain modalities, the findings could be considered incremental. Here are a few concerns.

Concerns:

  1. While it is likely difficult to recruit subjects for these studies, there should have control groups where the CS was room temperature water. The authors briefly discuss the potential confounds of repeated heat testing. These controls would have address this.
  2. Data showing individual PCES-amplitudes would have been informative.
  3. It is unclear why cortical potentials were not recorded with heat pain.
  4. The authors could add more discussion regarding the clinical relevance and/or use of this “new” (CS-PCES) paradigm. Specifically, what are the advantages of using this paradigm for disease diagnosis or neural circuit dissection. 

Reviewer 2 Report

The authors tackled an important issue concerning the methodology of CPM namely the differences in test protocols used to evaluate this concept in clinical practice. The manuscript is definitely of interest to readers, although I still have some concerns.

  • The main concern I still have is about the EEG analysis. Only a limited amount of information is provided on how these analysis were performed. Did the authors described the full pre-processing from their EEG analysis? Did the authors performed an ICA? What about blink artifacts, bad channels, noise, low and high pass filters? 
  • The authors focused on the N1 P1 peak amplitude. What was the time frame that was chosen to select these peaks? Were peaks automatically determined or did the authors used manual peak detection methods? 
  • Only EEG measurements were conducted with the electrical test stimulus. Why not with the temperature stimulus? More explanation concerning this issue is necessary in the manuscript. 
  • No figures are provided for the EEG analysis. 
  • Concerning the CPM protocol, the authors explained in the methodology that they use an absolute measure to determine the CPM effect. Why did the authors did not used a relative measure (i.e. dividing by baseline measurement)?
  • Previous studies already explored EEG measurements with electrical stimulation as test stimulus in healthy controls. These are not mentioned in the manuscript. Additionally, it not clear what the added value is of reporting EEG measurements for only one test stimulus. This does not add information about the comparison of both test stimuli. 

Round 2

Reviewer 1 Report

The authors have addressed my concerns.

Reviewer 2 Report

The authors adequately addressed my concerns. Concerning the reporting of CPM effects, Yarnitsky et al. reported the guidelines on how to conduct and report analysis in which they indicate that preferably both (absolute and relative) CPM effects are reported. By adding this to the table, CPM values are comparable to other studies.